

# Improvements of the OMI $O_2$-$O_2$ Operational Cloud Algorithm and Comparisons with Ground-Based Radar-Lidar Observations

J. Pepijn Veefkind[1,2], Johan F. de Haan[1], Maarten Sneep[1], Pieternel F. Levelt[1,2]

[1]Royal Netherlands Meteorological Institute (KNMI), De Bilt, The Netherlands
[2]Delft University of Technology, Faculty of Civil Engineering and Geosciences, Delft, The Netherlands

**Abstract**. The OMI (Ozone Monitoring Instrument) OMCLDO2 cloud product supports trace gas retrievals of for example ozone and nitrogen dioxide. The OMCLDO2 algorithm derives the effective cloud fraction and effective cloud pressure using a DOAS fit of the $O_2$-$O_2$ absorption feature around 477 nm. A new version of the OMI OMCLDO2 cloud product is presented that contains several improvements, of which the introduction of a temperature correction on the $O_2$-$O_2$ slant columns and the updated look-up-tables have the largest impact. Whereas the differences in the cloud fraction are limited to approximately 0.1, the differences of the cloud pressure can be up to 200 hPa, especially at cloud fractions below 0.3. As expected, the temperature correction depends on latitude and season. The updated look-up tables have a systematic effect on the cloud pressure at low cloud fractions. The improvements at low cloud fractions are very important for the retrieval of trace gases in the lower troposphere, for example for nitrogen dioxide and formaldehyde. The cloud pressure retrievals of the improved algorithm are compared with ground-based radar-lidar observations for three sites in the mid-latitudes. For low clouds that have a limited vertical extent the comparison is favorable. For higher clouds, which are vertically extensive and often contain several layers, the satellite retrievals give a lower cloud-height. For high clouds mixed results are obtained.

## Introduction

The Ozone Monitoring Instrument (OMI) is a imaging spectrometer developed by The Netherlands and Finland that has been launched in 2004 on board of the NASA EOS Aura satellite (Levelt et al., 2006). OMI has a continuous spectral coverage from 270-500 nm, with a resolution of approximately 0.5 nm. The primary data products from OMI are concentrations of trace gases, including ozone, nitrogen dioxide and formaldehyde. The trace gas retrieval algorithms rely on a priori information of cloud properties. For tropospheric trace gas retrievals, clouds are among the leading error sources in the retrieval (e.g. Boersma et al., 2011).

The OMI $O_2$-$O_2$ cloud product (OMCLDO2) contains information on the cloud fraction and cloud pressure for each ground pixel. The OMCLDO2 product has been designed to support the trace gas retrieval algorithms and is therefore driven by what these algorithms need for cloud information. The trace gas retrieval algorithms use the independent pixel approximation (Zuidema and Evans, 1998) representing clouds as Lambertian reflectors with a fixed albedo of 0.8 (Stammes et al., 2008). To be consistent with the trace gas retrievals, the OMCLDO2 product uses the same cloud model. The initial OMCLDO2 algorithm has been described by Acarreta et al. (2004). Because the amount of information





in the OMI spectral range is limited, the algorithm derives an effective cloud fraction and an effective
cloud pressure. The cloud fraction and cloud pressure are derived from the continuum radiance
and the depth of the $O_2$-$O_2$ absorption feature around 477 nm. The algorithm does not distinguish
between clouds and aerosols. Cloud-free conditions with significant thick aerosols layers will be
represented by small cloud fractions. Similarly, thin clouds, for instance cirrus, will also be represented
by a small cloud fraction. The main a-priori information that is used is the surface reflectance and the
surface altitude, which are obtained from static look-up tables. Validation studies (Sneep et al., 2008)
have shown that the effective cloud fraction compares well with effective cloud fractions derived from
the cloud optical thickness observed by MODIS (Moderate Resolution Imaging Spectroradiometer) and
that the derived cloud pressure determines a level somewhere near the middle of the clouds. This is
different from the cloud pressures derived from the thermal infrared, which are very sensitive near the
actual cloud tops. The OMCLDO2 retrieval is very similar to the FRESCO algorithm (Wang et al., 2008)
with the difference that it is based on $O_2$-$O_2$ rather than $O_2$ absorption lines. The reason for using $O_2$-$O_2$
is that the OMI spectral range doesn't cover the oxygen absorption bands. An important difference of
using the oxygen dimer is that its absorption scales with the oxygen density squared, which makes it
increasingly more sensitive to the lower altitudes in the atmosphere. Besides the OMCLDO2 algorithm,
there is also an OMI product based on the information from Raman scattering (Joiner et al., 2012; Joiner
and Vassilkov, 2006). It has been demonstrated that this product is also sensitive to the middle of cloud
layers, which has been referred to as the optical centroid pressure.
This paper describes version 2.0 of the OMCLDO2 product. Whereas updates and reprocessing was
performed regularly in the past, the version 2.0 contains the following improvements and extensions:
1.   A temperature correction is implemented which is needed because of the density-squared nature

60          of the $O_2$-$O_2$ absorption;

2.   Besides the independent pixel approximation, a second cloud model is implemented, which

62          represents the scene as a Lambertian surface at a certain pressure level. The retrieved parameters

63          are the scene albedo and scene pressure;

3.   The look-up-tables that are used to derive the cloud fraction and pressure have a higher number

65          of nodes, especially for the surface albedo and the surface altitude;

4.   A method has been implemented to remove outliers from the spectral fitting;
5.   The resolution of the a priori surface altitude is brought in line with the average OMI spatial

68          resolution;

6.   The gas absorption cross-sections are made consistent with the OMI $NO_2$ retrieval algorithm

70          (Geffen et al., 2014).

This paper is organized as follows: in section 2 we describe the OMCLDO2 algorithm, focusing on the
improvements that have been introduced in this version. In section 3 we discuss differences compared to
the previous version. In section 4 we present comparisons of the cloud pressure to ground based radar
observations.





**Algorithm** 75

The OMCLDO2 retrieval consists of two main steps: first a DOAS (Differential Absorption 76
Spectroscopty) fit is performed on the spectral region between 460 and 490 nm to derive the $O_2.O_2$ slant 77
column amount $N_{s,O2\text{-}O2}$ and the continuum reflectance $R_c$. In the second step these parameters are 78
converted into cloud fraction $c_f$, cloud pressure $p_{cld}$, scene albedo $A_{scn}$ and scene pressure $p_{scn}$. 79

**DOAS fit** 80

The DOAS fit is performed on the Earth's reflectance. OMI measures the Earth's radiance and once per 81
day the solar irradiance. The wavelength grids of the Earth radiance and solar irradiance differ, because 82
of the Doppler shift and because of non-homogeneous filling of the slit for partly cloudy scenes (Voors 83
et al., 2006). For each ground pixel, the radiance ($I$) and irradiance ($F$) are brought on the same spectral 84
grid (see Van Geffen et al., 2015) and the reflectance is calculated as $R(\lambda) = \frac{\pi I(\lambda)}{\cos\theta_0 F(\lambda)}$, where $\lambda$ is the 85
wavelength and $\theta_0$ is the solar zenith angle. Next, the following equation is used for the DOAS fit: 86

$$R(\lambda) = P(\lambda)e^{-(N_{s,O2O2}\sigma_{O2O2}(\lambda) + N_{s,O3}\sigma_{O3}(\lambda))} \cdot (1 + c_R \frac{I_R(\lambda)}{F(\lambda)}) \qquad (1)$$  87

where $P(\lambda)$ a polynomial of the first order, $N_{s,O2O2}$ the slant column of $O_2.O_2$, $\sigma_{O2O2}(\lambda)$ the $O_2.O_2$ cross 88
section convolved with the OMI slit function, $N_{s,O3}$ the slant column of $O_3$, $\sigma_{O3}(\lambda)$ the $O_3$ cross section 89
convolved with the OMI slit function, $I_R(\lambda)$ a synthetic radiance Raman spectrum convolved with the 90
OMI slit function and $c_R$ a scale parameter for the amount of Raman scattering. For the reference cross 91
sections for $O_2$-$O_2$ we use (Thalman and Volkamer, 2013) at 293 K and for $O_3$ we use (Bogumil et al., 92
2000) at 220 K. 93

We solve Eq. 1 using a modified Levenberg-Marquardt method, using the errors for the radiance and 94
irradiance as weights. The fit parameters are the slant columns $N_{s,O2O2}$ and $N_{s,O3}$, and $c_R$, and the 95
coefficients for the polynomial $P(\lambda)$. In addition, also the diagnostics of the fit is obtained, including the 96
residuals and error estimates for all fit parameters. The residuals are analyzed for possible outliers. Such 97
outliers may be caused by high-energy particles hitting the detector or by varying dark current. Although 98
al the information in the OMI Level 1B product is used to remove bad spectral pixels, some may remain. 99
For outlier detection several methods have been used (e.g. Richter et al., 2011), which are mostly based 100
on Gaussian statistics, i.e. by using the mean and standard deviation of the residual. Because particle hits 101
will cause only increases in detected radiance and because the mean and standard deviation themselves 102
are strongly affected by outliers, we selected the so-called box-plot method for outlier detection 103
(http://www.itl.nist.gov/div898/handbook/prc/section1/prc16.htm). This method determines lower and 104
upper values based on the 25[th] and 75[th] percentile of a distribution. If the lower quartile is $Q1$ and the 105
upper quartile is $Q3$, then the difference $(Q3 - Q1)$ is called the interquartile range or IQ. We define 106
outliers as those values smaller than $Q1 - 1.5\ IQ$ or larger than $Q3 + 1.5\ IQ$. After removal of the outliers, 107
we redo the fitting of the spectrum to provide the final fit parameters. We have noted that the outlier 108
removal is not stable; continuing iterating and each time applying the outlier removal procedure will 109
often result in more and more removed spectral pixels. We therefore iterate only one time, thus removing 110
the largest outliers. 111

112





**Conversion to Cloud and Scene Parameters**

**Radiative transfer modelling**

For the conversion of the DOAS fit parameters into respectively cloud fraction and pressure, and scene albedo and scene pressure, we use radiative transfer modeling. The difference between the independent pixel approximation (IPA) (Zuidema and Evans, 1998) that determines the cloud fraction and pressure, and the Lambertian equivalent reflectance (LER) model that determines the scene albedo and pressure, is illustrated in Fig 1. It is noted that the clouds and the ground surface in the IPA model are treated as opaque Lambertian reflectors. Therefore, the name LER maybe somewhat confusing, but is used for consistency with the existing literature. For each ground pixel, both the IPA and LER method is applied. The IPA requires a-priori information on the surface reflectance and surface pressure. The clouds are represented as Lambertian reflectors with an albedo of 0.8. Different studies have found that this is an optimal choice for the purpose of cloud corrections in trace retrieval schemes (see (Stammes et al., 2008) and references therein). Using such a high albedo for the clouds will represent thin clouds covering the entire ground pixel as small cloud fraction. Thus, the cloud-free part will implicitly model the transmission of light through the cloud, which is otherwise absent in the Lambertian cloud model.

For very small cloud fractions the cloud pressure derived using the IPA will become undetermined. In case of surface albedo's close to 0.8, e.g. over snow and ice, the IPA retrieval for both cloud fraction and pressure will become undetermined. In such cases, the LER method may be a good fallback.

For both the IPA and LER model, we use the same set of forward model simulations of the reflectance between 460 and 490 nm, see Table 1. These simulations are performed for a mid-latitude summer standard atmosphere. The correction for different temperature profiles is discussed later on in this section. On the simulated reflectance the same DOAS fit is performed as for the measured OMI spectra (Eq. 1). For all the nodes listed in Table 1, we obtain the slant column $O_2$.$O_2$ as well as the continuum reflectance at 475 nm. The continuum is computed by evaluating the polynomial $P(\lambda)$ for this wavelength.

**Look-up-table inversion**

Although we now have the information needed to derive the cloud fraction/pressure and the scene albedo/pressure, we invert the tables to improve the computational speed. Instead of having the cloud fraction and cloud pressure as nodes of the tables, we want to have the slant column $O_2$.$O_2$ and continuum reflectance as nodes. This conversion process involves interpolation and extrapolation, for which we use linear radial basis functions (http://docs.scipy.org/doc/scipy-0.15.1/reference/generated/scipy.interpolate.Rbf.html).

Because the simulated spectra cover a very wide range of conditions, it is unlikely that the extrapolations in this inversion procedure have a large effect on the final result. The inversion is illustrated in Fig. 2. The final result of the inversion procedure are look-up tables (LUTs) for the cloud fraction, cloud pressure, scene albedo and scene pressure on the nodes listed in Table 2. In the retrieval algorithm linear interpolation is applied on all dimensions, except for the solar zenith angle, for which spline interpolation is applied. This is implemented because of the non-linear behavior at large solar zenith angles.





**Temperature correction**
As will be described in this section, the slant column amount of $O_2$-$O_2$ depends on the temperature profile,
even if the cross section is not temperature dependent. This is due to the nature of the dimers, of which
the absorption scales with the pressure-squared instead of being linear with pressure. Because this effect
turns out to be significant, we have developed a temperature correction. This correction allows the use
of the LUTs described above, which have been derived for a single pressure-temperature profile. By
applying temperature correction, the $O_2$-$O_2$ slant columns are scaled to the values for the reference
temperature profile that has been used to construct the LUTs.
To understand the temperature effect of the $O_2$-$O_2$ slant columns, we write the reflectance as:

$$R(\lambda) = R_0(\lambda) \exp\left(-\int_{z_0}^{TOA} m(z,\lambda)\, n_{O2}^2\, \sigma_{O2-O2}(\lambda)\, dz\right), \qquad (2)$$


where $R_0(\lambda)$ is the reflectance if absorption by $O_2$-$O_2$ is ignored; $z_0$ is the altitude of a Lambertian cloud
or the Earth surface; $TOA$ is he top of the atmosphere; $m(z,\lambda)$ is the altitude resolved air mass factor
which is weakly wavelength dependent; $n_{O_2}(z)$ is the number density of oxygen and $\sigma_{O2-O2}(\lambda)$ is the
absorption cross section of $O_2$-$O_2$.
In hydrostatic equilibrium, the integral over the altitude can be replaced by an integral over the pressure,
using $dp/dz=-\rho(z)g$, where is the density of air. By expressing the density of air as $\rho(z)=M\,p\,/\,(R_g\,T(z))$,
where $M$ is the mean molecular mass of dry air and $R_g$ is the gas constant, Eq. 2 becomes:

$$R(\lambda) = R_0(\lambda) \exp\left(\int_{p_0}^{p_{TOA}} \frac{R_g}{M\,g}\, T(p)\, m(p,\lambda)\, n_{O2}^2(p)\, \sigma_{O2-O2}(\lambda)\, \frac{dp}{p}\right). \qquad (3)$$


Finally, we can express the number density of air in as $n_{O_2} = 0.21\, p/(k_b\, T(p))$, where $k_b$ is Boltzmann's
constant and we assume a mixing ratio of oxygen of 21%. Substituting this in Eq. 3 gives:

$$R(\lambda) = R_0(\lambda) \exp\left((0.21)^2\, \frac{R_g}{M\,g\,k_B^2}\, \sigma_{O2-O2}(\lambda) \int_{p_0}^{p_{TOA}} m(p,\lambda)\, \sigma_{O2-O2}(\lambda)\, \frac{p}{T(p)}\, dp\right), \qquad (4)$$


which shows that the reflectance and hence the slant column of $O_2$-$O_2$ changes when the temperature
profile changes. It is noted that this is due to the density-squared nature of the absorption of $O_2$-$O_2$. For
"normal" absorbers (no collision complex) the slant column is independent of the temperature profile,
apart from temperature dependence of the absorption cross section.

In order to investigate the magnitude of the bias that is introduced if the temperature dependence is
ignored simulations of the retrieval were performed. In the retrieval the mid-latitude summer profile is
used while for the simulations either a mid-latitude winter profile or a sub-arctic winter profile is used.
The bias was calculated for different true pressure levels of the cloud and for different cloud fractions.
Fig 3 shows that the maximum bias in the retrieved cloud pressure ranges from less than 50 hPa at large





cloud fractions to 200 hPa at very small cloud fractions. Such biases will have a significant impact on
trace gas retrievals, which are commonly limited to scenes with small cloud fractions.

The OMCLDO2 retrieval is based on a LUT approach and generating LUTs for different temperature
profiles in not feasible. Therefore we introduce a correction factor $\gamma$ that translates the measured slant
column into the slant column for the reference pressure-temperature profile. Using Eq 4., we can compute
$\gamma$ as:
$$\gamma = \frac{N_s^{ref}}{N_s^{meas}} = \frac{\int\limits_{p_c}^{p_{TOA}} m(p,\lambda)\frac{p}{T_{ref}(p)}dp}{\int\limits_{p_c}^{p_{TOA}} m(p,\lambda)\frac{p}{T(p)}dp} \qquad (5)$$

where $T(p)$ is the actual temperature profile taken and $T_{ref}(p)$ is the temperature profile used in the
creation of the look-up tables. In case of partial cloud cover and weak absorption we obtain
$$\gamma = \frac{N_s^{ref}}{N_s^{meas}} = \frac{(1-c_f)R_{clr}\int\limits_{p_s}^{p_{TOA}} m_{clr}(p,\lambda)\frac{p}{T_{ref}(p)}dp + c_f R_{cld}\int\limits_{p_c}^{p_{TOA}} m_{cld}(p,\lambda)\frac{p}{T_{ref}(p)}dp}{(1-c_f)R_{clr}\int\limits_{p_s}^{p_{TOA}} m_{clr}(p,\lambda)\frac{p}{T(p)}dp + c_f R_{cld}\int\limits_{p_c}^{p_{TOA}} m_{cld}(p,\lambda)\frac{p}{T(p)}dp} \qquad (6)$$

where $R$ is the reflectance at a representative wavelength in the fit window, $p_s$ is the surface pressure and
$p_s$ the cloud pressure, and the subscripts *clr* and *cld* refer to the clear part and the cloudy part of the pixel,
respectively.
To implement the temperature correction factor, new look-up-tables for the $O_2$-$O_2$ air mass factors $m(p,\lambda)$
and the corresponding reflectance for a wavelength in the middle of the fit window have been generated.
In the retrieval algorithm, the temperature correction is applied in an iterative manner because the cloud
fraction and pressure should be known to compute $\gamma$. As a default, we use three iterations to compute $\gamma$.
**A-priori information**
The OMCLDO2 version 2 uses the following *a-priori* information.
For the absorption cross-sections for $O_2$-$O_2$, ozone and optionally $NO_2$, as well as for the radiance Raman
scattering, we use the spectra described in Van Geffen et al., (2015).  For the surface reflectance, the
OMI derived monthly mean database described in Kleipool et al., (2008) extended to 5 years of OMI
data is used. For the temperature profiles needed for the temperature correction, we use a monthly mean
climatology at four times per day (00, 06, 12 and 18 UTC), derived from the NCEP reanalysis data for
the period 2005-2014. Actual temperatures maybe somewhat better than using a climatology. However
for practical reasons related to the operational data processing facility, we have decided to use a
temperature climatology. For detecting snow and sea-ice coverage, the Near-real-time Ice and Snow
Extent (NISE) product (Nolin et al., 1998) is used.





**Impact of algorithm updates**

In this section we first compare the OMCLDO2 version 2 with the version 1.2.3 for one day of data. Next, the impacts of each of the improvements are discussed separately. The impact of the improvements are summarized in Table 2.

Figure 4 shows the OMCLDO2 retrieval results for 14 May 2005. This day has been selected arbitrarily from the OMI data record. Note that we also have analysed other days, which show consistent results. Figure 4a and b show the effective cloud fraction and the effective cloud pressure. Figures 4c and d show the difference between version 2 and version 1.2.3. For areas with low effective cloud fractions, the effective cloud fraction is approximately 0.01 higher in the version 2. Over the high latitudes in the northern hemisphere considerably large positive and negative differences occur. These occur over snow and ice, where the retrieval algorithm has problems to distinguish the clouds from the highly reflective surface. Under such conditions, the accuracy of the retrieved effective cloud fraction will be very low. Due to the assumed cloud albedo of 0.8, the cloud fraction will become undetermined when the surface albedo is also close to this value.

The differences in effective cloud pressure are shown in Fig 4d. Version 2 shows higher cloud pressure in the tropics and sub-tropics, and lower cloud pressures and mid and high latitudes. As discussed below, this zonally dependent effect is caused by the temperature correction introduced in version 2. Especially in the tropics, the differences in the cloud pressures are largest in regions with low cloud fractions. Overall the uncertainty in the cloud pressure retrievals is a strong function of the effective cloud fraction. This is illustrated in Figure 5, which shows the precision of the effective cloud pressure retrievals as a function of the effective cloud pressure. The precision is calculated by the propagation of the DOAS fit errors of the $O_2$-$O_2$ slant columns and of the continuum reflectance. For cloud fractions below 0.1 the average precision is larger than 20 hPa with a very large spread, whereas for cloud fractions above 0.9 the precision is less than 10 hPa with a much smaller spread. It is noted that other errors sources, for example in the *a priori* surface albedo will also have a much stronger impact at low effective cloud fractions.

**Temperature correction**

The correction for the temperature dependence is described above. Based on a temperature climatology, a correction factor is computed and applied to the $O_2$-$O_2$ slant columns. Figure 4g shows the temperature correction factor for the OMI observations on 14 May 2005. Because the temperature correction factor is computed relative to the midlatitude summer atmosphere, it is larger than 1 in the tropics and smaller at the higher latitudes. On top of this general behavior there is spatial structure related to cloud structures, especially when the clouds are at high altitudes and have significant optical thickness. The effect of clouds on the temperature correction factor is described in Eq. 6. For high and thick clouds the temperature correction is in most cases closer to 1, indicating that the largest differences between the climatological temperature and the mid-latitude summer atmosphere occurs at the lowest altitudes.

To test the impact of the temperature correction factor on the effective cloud fraction and pressure, we produced datasets with and without the temperature correction applied for two days of OMI data in different seasons (14 May 2005 and 15 November 2005). While the impact on the cloud fraction is



negligible, the impact on the cloud pressure can be significant. Fig. 6 shows the difference between the
retrievals without and with the correction applied, as a function of the effective cloud fraction. The impact
of the correction on the cloud pressure increases towards smaller cloud fractions. Depending on whether
the correction factor is smaller or larger than 1, the impact on the cloud pressure can be both positive or
negative. For cloud fractions below 0.2, the impact of the temperature correction can be as large as -100
to 150 hPa, whereas for cloud fractions larger than 0.2 the impact is in the range -20 to 40 hPa. For the
higher latitudes ($\gamma>1$) the clouds are at lower pressures (higher altitude) when the temperature correction
is applied, whereas in the tropics and sub-tropics the effects is reversed.
Fig. 6 can be compared to Fig. 2, which is based on retrieval simulations. Although Fig. 6 shows the
difference with and without the temperature corrections, and Fig. 2 shows the difference with the
simulated truth, the behavior and magnitude of the bias is very similar. It is noted that for Fig. 2 only
temperature profiles have been used which are colder in the troposphere than the reference mid-latitude
summer atmosphere. Therefore, Fig 2 shows only positive biases, whereas in the tropics and sub-tropics
Fig. 6 also shows negative values.
**Look-up-tables**
To test the impact of the LUTs that are used to derive the effective cloud fraction and effective cloud
pressure, we produced a datasets using the version 2 algorithm with the new and the old LUTs. The cloud
fraction with the new LUTs is about 0.01 larger than with the old version, except over snow and ice
regions where the cloud fraction is in most cases significantly smaller. Because over snow and ice
covered regions the cloud fraction is highly uncertain as the algorithm is not able to distinguish clouds
from highly reflective surfaces, this impact is not unexpected.
The effect of the new LUTs on the effective cloud pressure is shown in Fig. 7c. This figure shows the
difference in the cloud pressure (old minus new) as a function of the effective cloud pressure. The
differences become significant at cloud fractions smaller than 0.25, where the difference shows an
oscillating behavior. At a $c_f$ of approximately 0.125 a minimum is reached and at smaller cloud fractions
the mean difference reverses sign and increases towards lower $c_f$. To investigate the nature of this
behavior, Fig. 7a and 7b show the distribution of the retrieved cloud pressures as a function of cloud
fraction for the old and new LUT datasets. From these figures it is clear that the origin of the oscillating
behavior of the difference is in the retrievals with the old LUTs. Fig 7a shows that with the old LUTs the
cloud pressure increasing strongly towards lower cloud fractions, for which we have no physical
explanation. The results with the new LUTs (Fig. 7b) do not show this. We attribute the large
improvements with the new LUTs to the larger number of radiative transfer calculations on which it is
based, as well as the improved interpolation scheme that was used to produce it.
Figs. 7a and 7b also show that the effective cloud pressure for the largest $c_f$ bin is significantly larger. A
further inspection showed that this is caused retrievals over snow and ice covered regions, for which the
cloud pressure retrievals are highly uncertain. For such cases the scene albedo and pressure provided by
the version 2 algorithm can be used.



**Outlier removal**

The outlier removal procedure that was introduced in the version 2 of the algorithm removes spectral pixels from the DOAS fit after evaluation of the fitting residuals. Outliers can have different behavior: they can be transient, e.g. occurring only for spectral pixels for a few pixels, or they can occur systematically for certain spectral pixels. When outliers are detected they are removed from the data, which will decrease the number of wavelengths used in the DOAS fit. Fig. 4h shows the number of wavelengths used in the fit for 14 May 2005. The most prominent feature are the reduced values over the South America caused by the South Atlantic Anomaly (SAA). In this region the number of high energetic particles hitting the OMI detectors is significantly increased (Dobber et al., 2006), resulting in spikes in the data. It is noted that also the Level 0-1B processor flags transient pixels, so Fig. 4h is the result of the Level 1B flags in combination with the outlier removal procedure. In addition to the SAA, figure 4h also shows stripes in the along-track direction, as well as features related to geophysical conditions (for example higher values of Australia and the India).

The impact of the outlier removal procedure was tested by running the algorithm with and without the procedure switched on for 14 May 2005. The differences in the retrieved effective cloud fraction are negligible, whereas the impact on the effective cloud pressure depends on the cloud fraction. The mean difference is not significant, but the standard deviation of the difference varies for 16 hPa for $c_f < 0.2$ to 3 hPa for $c_f < 0.8$.

We also inspected the root-mean-square error (RMSE) of the DOAS fit as a fit quality indicator. Although the difference in RMSE with and without the outlier removal did not differ significantly from zero, the distribution is skewed towards larger RMSE values when the outlier removal is switched off. This indicates that the outlier removal procedure improves the fit for cases with a high RMSE.

**Digital Elevation Model**

The version 2 of the algorithm uses a DEM with a resolution of approximately 20 km, which is closer to the spatial resolution of OMI compared to the 3 km resolution DEM used in previous versions. The 20 km resolution DEM is constructed from the Global Multi-resolution Terrain Elevation Data 2010 (Danielson and Gesch, 2011).

The impact of the new DEM will be largest in mountainous terrain. Fig. 8 illustrates the effect on the retrieved effective cloud pressures over Europe for 14 May 2005. This is the same day as shown in Fig. 4. Fig. 8a shows that significant impacts of the new DEM are restricted to the main mountain ranges. The difference between using the old and new DEM can be both positive and negative. The impact increases towards the lower cloud fractions, when more signal comes from the surface and an accurate knowledge of the surface altitude becomes more important. Fig 8b shows that for most pixels the impact is smaller than ±50 hPa.

**Cross sections**

In the new version of the algorithm, absorption cross-sections and the Raman radiance spectrum have been updated. The impact of this change was tested by running the algorithm with the old and the new cross sections. The impact on the cloud fraction was negligible. Using the new cross sections increased



the effective cloud fractions by 23±23 hPa. The difference in the root-mean-square error of the DOAS
was not significant. The new cross-sections didn't significantly reduce the residuals of the DOAS fit.
**Scene albedo and scene pressure**
As described in the algorithm section, for each ground pixel the scene albedo and scene pressure is
derived. The most important application of these parameters is over bright surfaces such as snow and
ice, where the surface albedo becomes close to the assumed cloud albedo of 0.8 and no meaningful cloud
fraction and pressure can be derived. Fig. 9 shows a comparison of the retrieved scene pressure with the
surface pressure derived from the DEM, assuming a sea level pressure of 1013 hPa. The figure shows a
very good agreement between the retrieved scene pressure and the DEM over Greenland. This figure
presents the comparison for the OMI cross track pixel 20, but other cross pixels show similar results. It
demonstrates the capabilities of the scene pressure for bright surfaces. Also, it is an indirect validation
of the retrieved $O_2$-$O_2$ slant columns. A correction of the $O_2$-$O_2$ slant columns, as is sometimes used in
ground based DOAS measurements (for a discussion see (Spinei et al., 2015)), is clearly not necessary
for the OMI retrievals.
Over dark surface, such as oceans, the scene pressure is less well understood. For some areas over the
ocean the retrieved scene pressure is significantly larger than the sea level pressure. Therefore, we
recommend using the scene albedo and scene pressure only for ground pixels which are covered with
snow and/or ice.
**Comparison with ground-based radar**
The changes made in the version 2 of the OMCLDO2 algorithm have a stronger impact on the cloud
pressure retrieval then on the cloud fraction retrieval. Therefore, we focus in this section on comparisons
of the cloud pressure retrievals with correlative data. Because of the use of the IPA cloud model (Fig. 1),
it is not straightforward to compare the retrieved cloud pressure to profile information on cloud
parameters. As discussed below, we compare the OMI retrievals with ground based radar data, for which
the sensitivity to cloud droplet size is very different; the OMI retrievals are sensitive to the optical
extinction with scales with droplet size to the power 2, whereas the radar reflectivity scales with droplet
size to the power 6. Thus, using these radar data it is not possible to compare the same quantity, which
is required in a validation study. Rather than a validation study, we focus on explaining the differences
between the OMI retrievals and the radar data, given their different sensitivities. This comparison uses a
similar approach as was used for comparing SCIAMACHY cloud products with radar data (Wang and
Stammes, 2014).
We present comparisons for three sites: Cabuw, The Netherlands, Lindenberg, Germany and the ARM
Southern Great Plains, U.S.A., for the period January to June 2006. These datasets were selected because
of the continuous data availability for these sites in the Cloudnet datatabase.



**Cloudnet data**

The Cloudnet dataset is the Level 2 classification product (Illingworth et al., 2007), which is available
approximately every 30 seconds. This product classifies each vertical layer as one of 11 classes, which
distinguish ice and water clouds, precipitation, aerosols, insects, clear sky and combinations thereof. We
attribute a value of 1 to layers that are classified as cloudy (classes 1-7) and 0 to layers identified as non-
cloudy. For profiles containing at least one cloudy layer, we compute the cloud mid-height as the average
of the altitude of the cloudy layers. Next we average all the profiles in the time window of +/- 30 minutes
of an OMI overpass. We also compute the average and standard deviation of the cloud mid-height over
this time window and determine for the average cloud profile if it is single-layer or multi-layer.
It is noted that this procedure for computing the cloud mid-height doesn't take the optical thickness of
the layers into account; a optically thick cloud and optically thin cloud are weighted the same in the cloud
mid-height. Weighting with the optical thickness - or even better, with the sensitivity of the $O_2$-$O_2$ cloud
algorithm- would make a comparison much more direct. Unfortunately, information on the full optical
thickness profile is not available from the Cloudnet data. Alternatively, we could use the radar reflectivity
as weighting parameter. However the radar reflectivity is very sensitive to cloud particle size, which is
also not a good representation for the cloud extinction in the visible. We therefore decided to use the
simple weighting described above. This weighting gives the same weight to optically thin cloud layer as
to optically thick layers, whereas the $O_2$-$O_2$ is cloud pressure retrieval is much more sensitive to the thick
layers.
Further filtering of the Cloudnet data was done using the following criteria:
• The standard deviation of the cloud-mid height should not exceed 1.5 km, to avoid cases with
large temporal variability during the OMI overpass;
• At least one layer in the profile should be cloudy during at least 50% of the time averaging
window.

**OMI collocated data**

For the OMI cloud data we average all the ground pixels of which the center is within 30 km distance of
the ground station. For these pixels we determine the mean and standard deviation for the cloud fraction
and pressure. We convert the cloud pressure to altitude using a scaling height of 8 km. We filter the OMI
data using the following criteria:
• The effective cloud fraction should exceed 0.2, because the cloud pressure for low cloud fraction
has a large uncertainty;
• The standard deviation of effective cloud pressures should not exceed 1.5 km, to exclude cases
with large horizontal variability.

**Results**

Figure 10 shows a comparison between the Cloudnet data and the OMI effective cloud pressure for the
collocations over Cabauw for the period January to June 2006. The cases presented in this figure are
ordered by increasing mid-height of ground-based data. The following regimes can be distinguished in
this data set:





1. Case 1-50: These are low level clouds with limited vertical extent. The OMI effective cloud height and the ground stations mid-height are in good agreement.
2. Case 51-129: According to the Cloudnet the majority of these cases consist of vertically extended, and often multi-layered cases. For these cases the OMI effective cloud height is generally lower than the ground station mid-height.
3. Case 130-135: Theses cases have high clouds with limited vertical extent. The OMI effective cloud height compares well, except for the outlier for cases 131. However, the number of collocations in this regime is small.

It is noted that the boundaries of these three regimes are not hard.

Figure 10 shows that for single layer clouds with a limited vertical extent, the $O_2$-$O_2$ effective cloud height and the Cloudnet derived mid-height are in agreement. This shows that the OMI derived product is capable of retrieving cloud height ranging from low clouds to high clouds. For vertically extended clouds, the OMI derived cloud heights are generally lower than the radar-lidar derived heights. A plausible explanation for this difference is that in these cases there are thin high clouds overlaying thicker low-level water clouds. Whereas the radar-lidar mid-heights have equal sensitivity, the $O2$-$O2$ cloud height will be more sensitive to the optically thick layers.

When we include not only Cabauw, but also Lindenberg and the ARM-SGP site, we get a similar picture. Figure 11 shows a comparison for all these sites for the period January-June 2006, where the single and multi-layer cloud cases are distinguished. Good correlation is observed for the cloud range of 0-2.5 km, where the single-cloud layers dominate. In the region between 2.5 and approximately 8 km the multi-layer clouds dominate and the $O_2$-$O_2$ cloud-height is lower than the radar-lidar cloud mid-height. Above 8 km we find both good comparison but also very large differences, although the number of points is very limited. As we are interested in the average comparisons, we did not investigate individual cases where big differences occurred.

**Conclusions**

We present a new version of the OMI OMCLDO2 Level 2 cloud product. This product is an important input for several of the operational OMI Level 1-2 algorithms. The new version contains six major improvements

1. The correction for the temperature sensitivity of the DOAS fit;
2. Improved look-up-tables for computing the effective cloud fraction and effective cloud pressure;
3. Retrieval of the scene pressure and scene albedo for every ground pixel, using the Lambertian Equivalent Reflector model;
4. Outlier removal procedure in the DOAS fit.
5. Updated of the reference cross sections;
6. Introduction of a DEM with a similar spatial resolution as the OMI ground pixels.

We show that the impact of these changes on the retrieved effective cloud fraction is for most ground pixels less than 0.01. The impact on the effective cloud pressure is larger: especially for cloud fractions less than approximately 0.3 the differences compared to the previous operational version can be as large





as 200 hPa. These differences are mainly caused by the temperature correction and the introduction of
the new look-up tables. Due to the temperature the differences have a latitudinal and seasonal dependent
behavior, where the updated algorithm gives higher cloud pressures at higher latitudes and lower
pressures in the tropics and sub-tropics. Also it was found that the new look-up-tables gives better results
at low cloud fractions.
Cloud pressure retrievals have been compared to ground based radar-lidar observations in Cabauw,
Lindenberg and the ARM-SGP site. It was found that for low clouds, up to approximately 2.5 km, the
satellite retrievals and ground-based results compare favorably. For clouds in the range between 2.5 and
approximately 8 km the ground-based observations indicate many multi-layer and vertically extensive
clouds. For these clouds the satellite retrieved cloud heights are generally lower, probably because the
algorithm is more sensitive to the optically thick low-level clouds. For high clouds (>8 km) mixed results
are found. The differences with the lidar-radar can be explained by the different sensitivity of the lidar-
radar observations versus the satellite observations.
We conclude that the new version of the OMCLDO2 product is a significant improvement of the previous
versions, especially for the cloud pressure at cloud fractions smaller than approximately 0.3. This is very
important for cloud corrections in retrievals of gases like nitrogen dioxide, sulphur dioxide and
formaldehyde, which are very sensitive to the cloud pressure.
After the reprocessing of the entire OMI data record, the stability of the product should be investigated,
and the scene pressure and scene albedo should be validated.

**Acknowledgements**
This work was funded by the Netherlands Space Office under the science OMI contract.
We acknowledge the EU Cloudnet and the ACTRIS projects for providing the cloud classification
dataset, using Radar-Lidar cloud classification products for the Cabauw, Lindenberg and the ATM-SGP
sites.

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





**Tables**

**Table 1: Nodes for the radiative transfer calculations. Note that cloud fractions smaller than 0 and larger**
**than 1 are included to enlarge the parameters space.**

| Parameter | Nodes |
|---|---|
| solar zenith angle [°] | 0.0, 9.3, 21.2, 32.9, 44.2, 54.9, 64.8, 73.5, 80.8, 86.1 |
| viewing zenith angle [°] | 0.0, 9.3, 21.2, 32.9, 44.2, 54.9, 64.8, 73.5 |
| relative azimuth angle [°] | 0, 30, 60, 90, 120, 150, 180 |
| surface albedo | 0.0, 0.01, 0.025, 0.05, 0.075, 0.1, 0.15, 0.2, 0.25, 0.325, 0.4, 0.5, 0.6, 0.7, 0.8, 0.9, 1.0 |
| surface/cloud pressure [hPa] | 1013, 963, 913, 863, 813, 763, 713, 663, 613, 563, 513, 463, 413, 363, 313, 263, 213, 163, 113, 63 |
| cloud fraction | -0.1, -0.05, 0., 0.01, 0.02, 0.04, 0.06, 0.08, 0.1, 0.125, 0.15, 0.175, 0.2, 0.25, 0.3, 0.35, 0.4, 0.45, 0.5, 0.55, 0.6, 0.65, 0.7, 0.75, 0.8, 0.85, 0.95, 1.0, 1.1, 1.2 |






**Table 2: Nodes for the continuum reflectance and the slant Column O2-O2, for the cloud fraction/pressure**
**and scene albedo/scene pressure look-up-tables. The solar zenith angle, viewing zenith angle, relative azimuth**
**angle, surface albedo and surface/cloud pressure nodes are the same as given in Table 1.**

| Parameter | Nodes |
|---|---|
| Continuum reflectance $R_c$ at 477 nm | 0.00, 0.05, 0.10, 0.15, 0.20, 0.25, 0.30, 0.35, 0.40, 0.45, 0.50, 0.55, 0.60, 0.65, 0.70, 0.75, 0.80, 0.85, 0.90, 0.95, 1.00, 1.05, 1.10, 1.15, 1.20, 1.25, 1.50, 1.75, 2.00 |
| Slant Column $O_2$-$O_2$ [$10^{44}$ $molec^2$ $cm^{-5}$] | 0.00, 0.05, 0.10, 0.15, 0.20, 0.25, 0.30, 0.35, 0.40, 0.45, 0.50, 0.55, 0.60, 0.65, 0.70, 0.75, 0.80, 0.85, 0.90, 0.95, 1.00, 1.10, 1.20 |









Table 3. Impact of the improvements of the effective cloud fraction and effective cloud pressure
retrievals.

| Improvement | Impact on $p_{cld}$ | Impact on $c_f$ |
|---|---|---|
| Temperature correction | Decreasing at higher latitudes<br>Increasing in the tropics and sub-tropics<br>$\Delta p_{cld}$ : -100 to 150 hPa for $c_f < 0.2$<br>$\Delta p_{cld}$ : -20 to 40 hPa for $c_f > 0.2$ | negligible |
| New look-up-tables | Impact is non-significant for $c_f > 0.3$<br>$\Delta p_{cld}$ : -60 to 220 hPa for $c_f < 0.3$ | $\Delta c_f$ : -0.01 except for high surface reflectivity for which $\Delta c_f > 0.05$ |
| Outlier removal | No systematic impact | negligible |
| DEM | Impact restricted to mountainous terrain.<br>$\Delta p_{cld}$ for most pixels smaller than +/- 25 hPa | negligible |
| Cross-sections | $\Delta p_{cld}$ : 23 +/- 23 hPa | negligible |







**Figures**

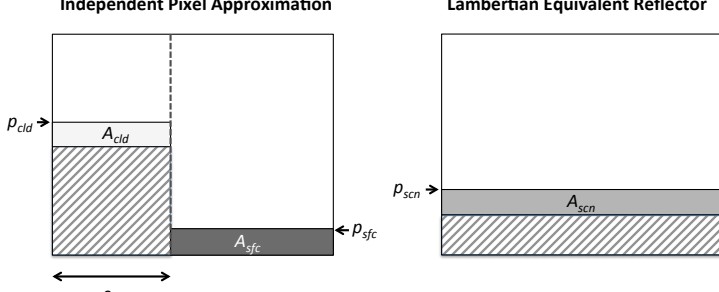


**Figure 1: The Independent Pixel Approximation versus the Lambertian Equivalent Reflector model. In the**
**IPA a ground pixel is modeled as the weighted sum of a cloudy part, (a Lambertian surface with an albedo of**
**$A_{cld}$ at a pressure level $p_{sfc}$) and a clear part (a Lambertian surface with an albedo of $A_{sfc}$ at a pressure level**
**$p_{sfc}$). The effective cloud fraction $c_f$ is used for the weighting of the cloudy and clear contributions. The IPA**
**method uses *a priori* information on $A_{sfc}$, $A_{cld}$ and $p_{sfc}$. In the LER model the ground pixel is modeled as a**
**Lambertian surface with a albedo $A_{scn}$ at a pressure level $p_{scn}$. The LER method doesn't rely on *a priori***
**information. Note that the hatched areas below the opaque Lambertian indicate that these regions do not**
**contribute in the radiative transfer calculations.**






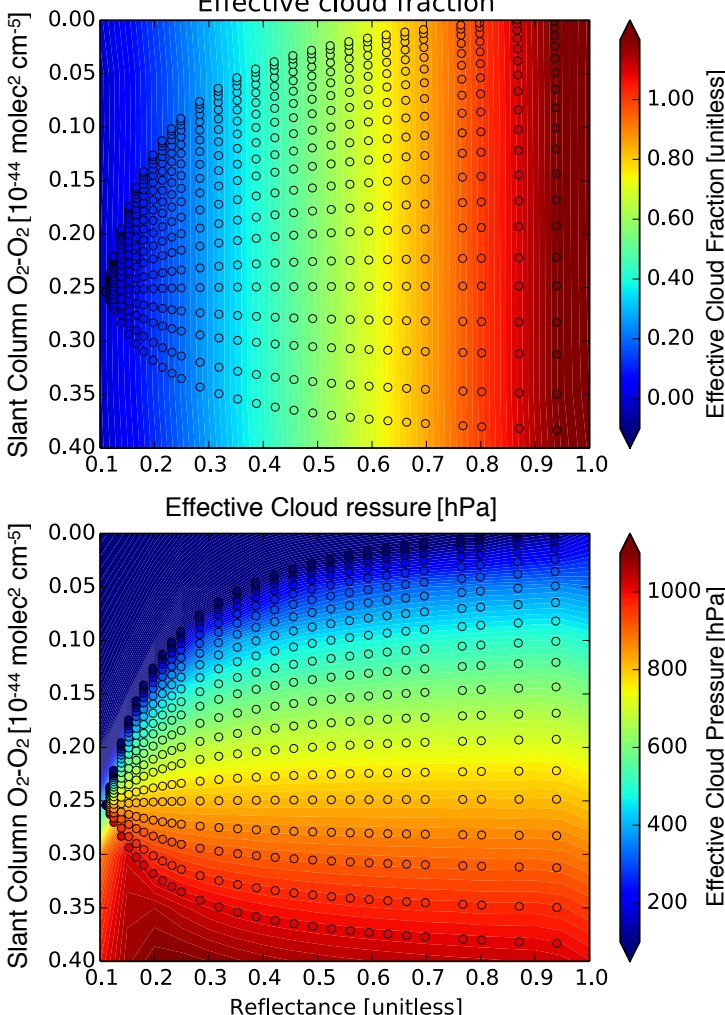


**Figure 2: Example of a slice of the effective cloud fraction LUT (top panel) and effective cloud pressure LUT (bottom panel), showing the LUT value as a function of the continuum reflectance $\rho_c$ and the slant column O$_2$-O$_2$ $N_{s,O2O2}$. The background colors show the values in the LUT derived from interpolation and extrapolation of the DOAS fit results, which are shown as the color-filled symbols. The other LUT nodes are fixed to the following values: solar zenith angle 44.2°; viewing zenith angle 21.2°; relative azimuth angle 0.0°; surface albedo 0.05; surface altitude 0 m.**


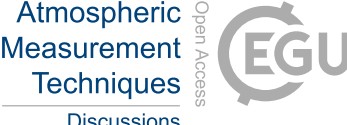



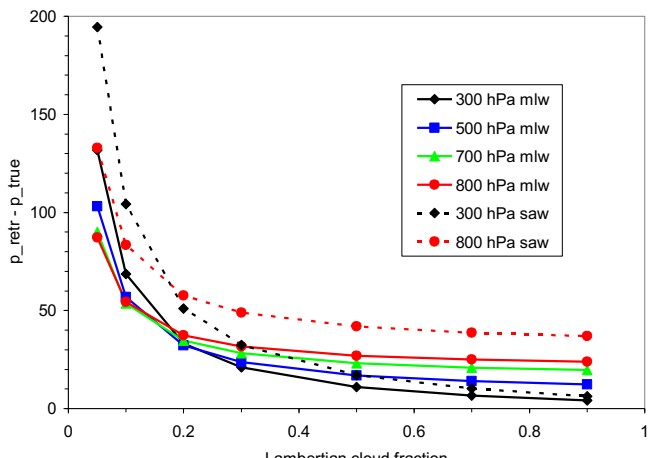


**Figure 3: Bias in the retrieved pressure ($p_{retr} − p_{true}$) in hPa when in the retrieval a mid-latitude summer**
**temperature profile is used whereas in the simulation a mid-latitude winter profile (mlw) or a sub-arctic**
**winter profile (saw) is used. The results are plotted as a function of the cloud fraction and for different**
**pressure levels of the cloud used in the simulation. The surface albedo is fixed at 0.05, the cloud albedo is 0.80,**
**the solar zenith angle is 60 degrees and the viewing direction is nadir.**





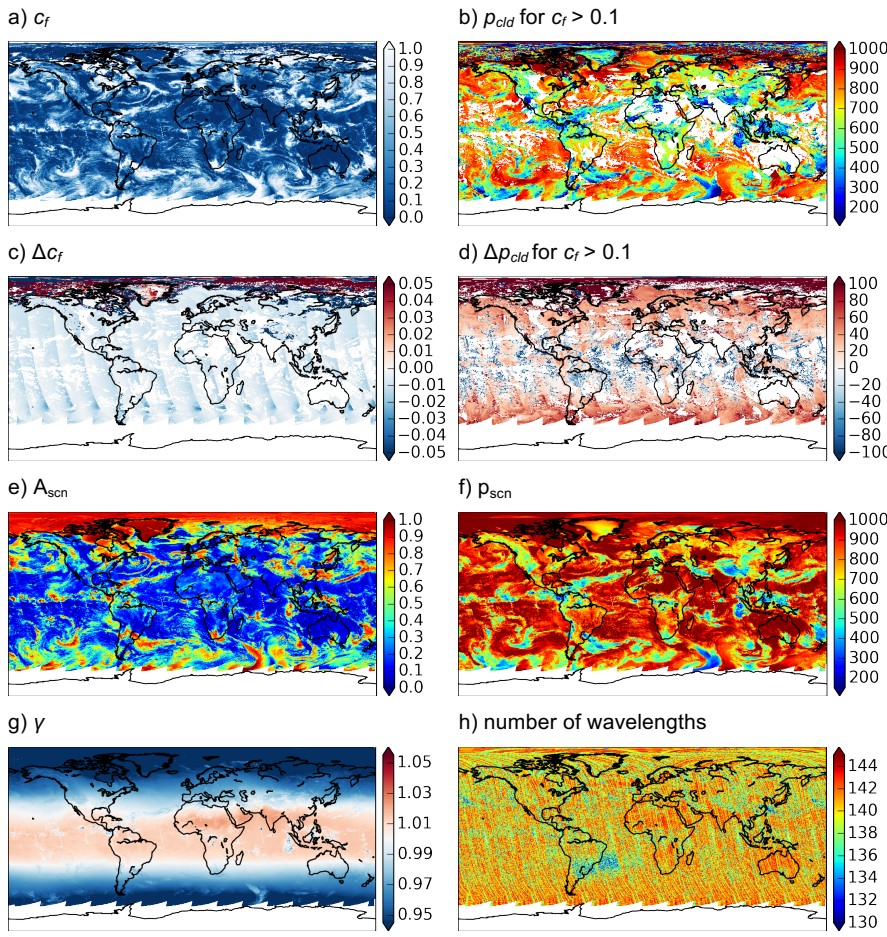

Figure 4: Results from the OMCLDO2 version 2 algorithm for 14 May 2005. a) effective cloud fraction, b)
effective cloud pressure, c) difference of the effective cloud fraction (version 1.2.3 minus version 2), d)
difference of the effective cloud pressure (version 1.2.3 minus version 2), e) scene albedo, f) scene pressure, g)
SCD temperature correction factor γ, and h) number of wavelengths used in the DOAS fit.





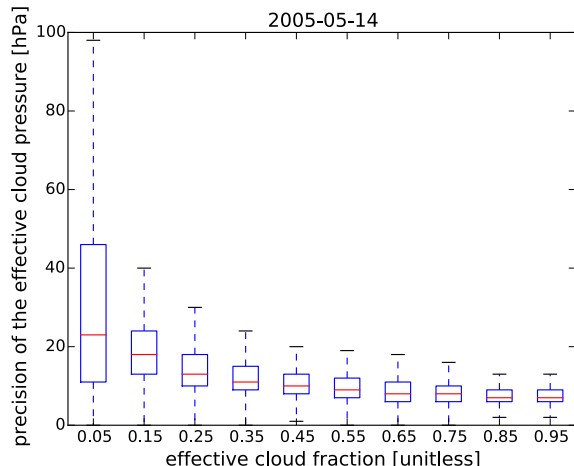


**Figure 5: Box-whisker plot of the precision of the effective cloud pressure as a function of the effective cloud fraction for 14 May 2005.**



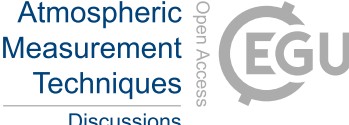




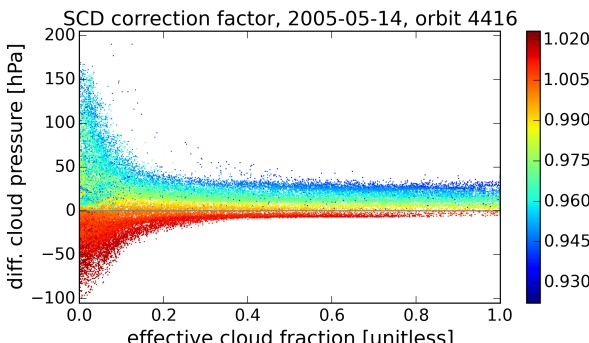


**Figure 6: Difference in the effective cloud pressure due to the temperature correction (without correction minus with correction) plotted as function of the effective cloud fraction. The colors of the symbols indicate the SCD correction factor.**




a) $p_{cld}$ old

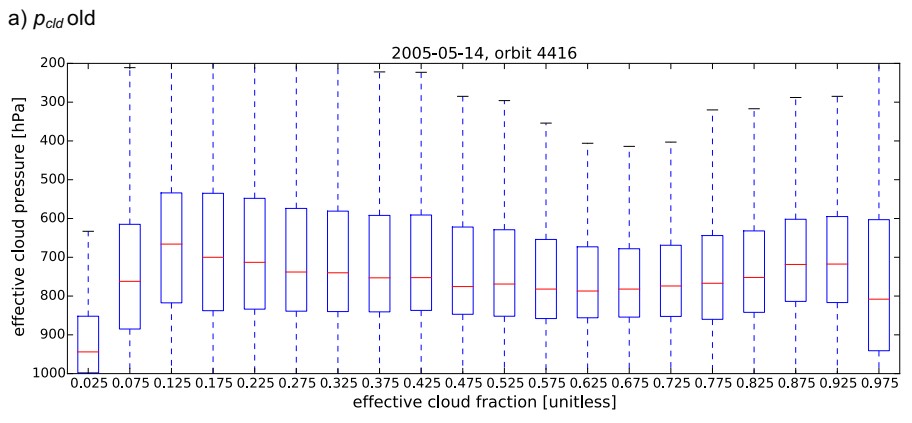

b) $p_{cld}$ new

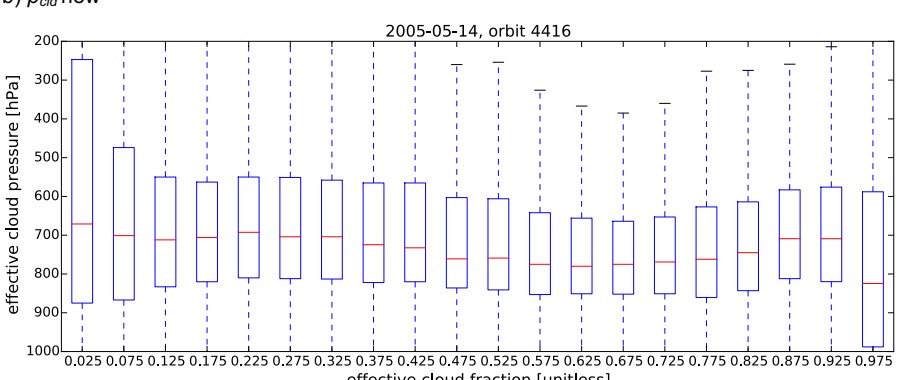

c) $\Delta p_{cld}$ old - new

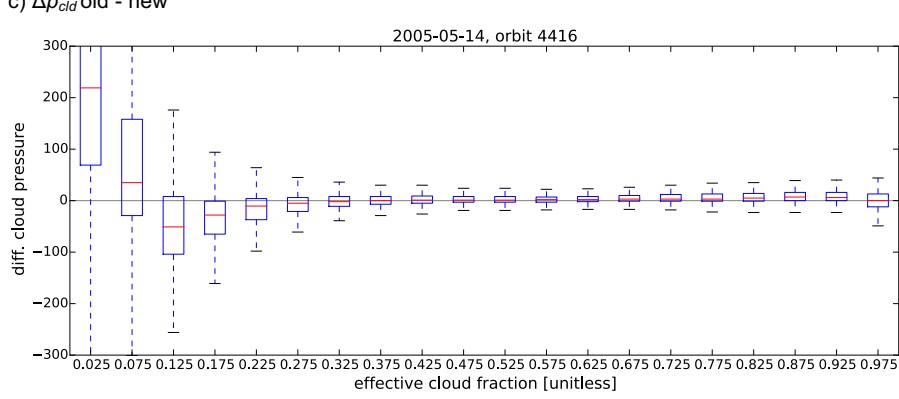

**Figure 7: Box-whisker plots of the effective cloud pressure as a function of the effective cloud fraction. The top plot is for the old LUTs, the middle for the new LUTs and the bottom plot for the difference of old minus new.**

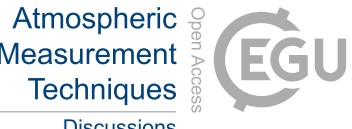

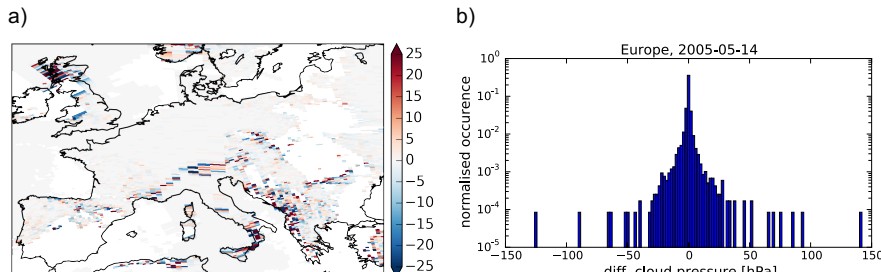


**Figure 8: Difference in the effective cloud pressure (old DEM minus new DEM) for effective cloud fractions exceeding 0.1 over Europe for 14 May 2005. Left panel: map of the differences over Europe, right panel: histogram of the differences over Europe on a logarithmic scale.**







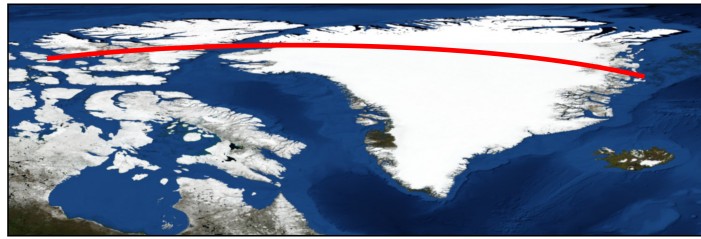

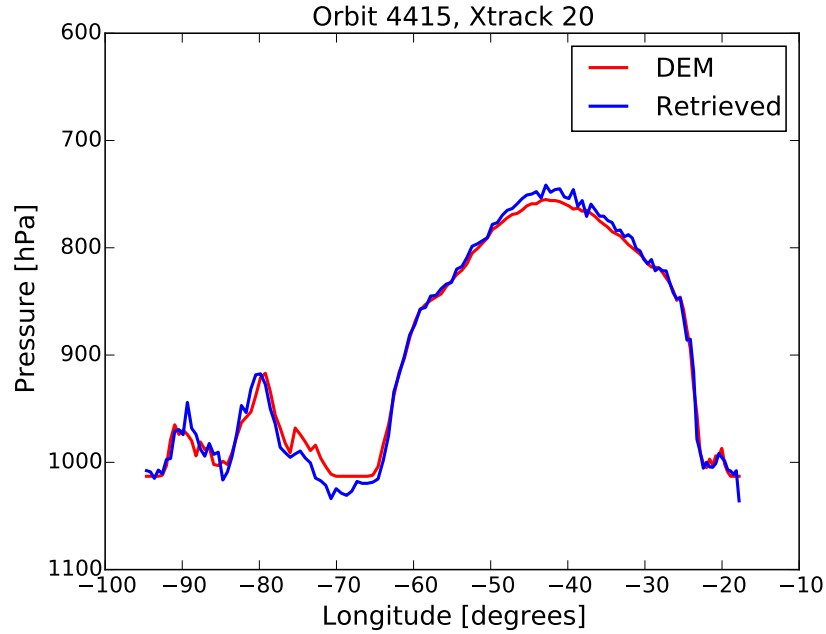

Fig 9: Top panel: map of the position of the ground pixels centers. Bottom panel: comparison of the retrieved scene pressure and the surface pressure derived from the DEM, plotted as a function of the longitude.








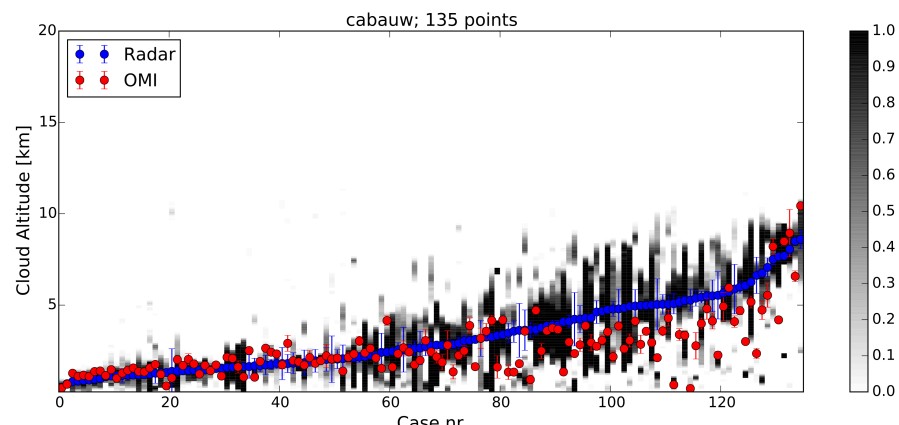


**Figure 10: The effective cloud altitude retrieved from OMI (red), compared to radar cloud information for Cabauw (blue), The Netherlands. The grey background is the vertically resolved cloud occurrence derived from the radar data for the period +/- 30 minutes of the OMI overpass. The cases are ordered according to the ground station cloud mid-height.**

614





615

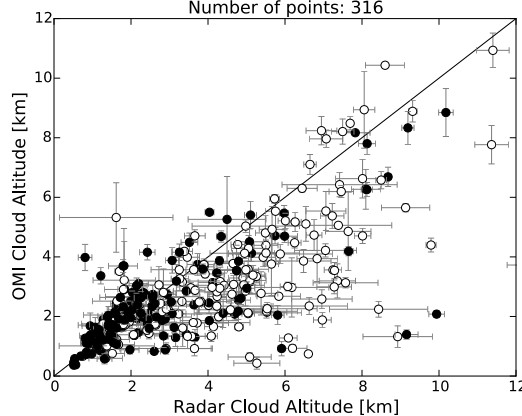

616

**Figure 11: The retrieved effective cloud altitude from OMI, plotted as a function of the radar derived cloud altitude. Closed symbols are for single-layer clouds, open symbols for multi-layer clouds.**