# Peer review of "Improvements of the OMI $O_2$-$O_2$ Operational Cloud Algorithm and Comparisons with Ground-Based Radar-Lidar Observations"

_Atmospheric Measurement Techniques, 2016_

## Referee Comment (RC1) · Anonymous Referee #1 · 4 May 2016

Review of "Improvements of the OMI O2-O2 Operational Cloud Algorithm and Comparisons with Ground-Based Radar- Lidar Observations" by Veefkind et al.

General comments

This paper describes a number of improvements of the OMI operational O2-O2 cloud algorithm which provides important input data for OMI trace gas algorithms. The authors carry out comparisons of the new version of cloud products with the old version. They also compare the retrieved effective cloud pressures with ground-based radar data. The authors conclude that while the impact of the improvements on effective cloud fraction is small the impact on effective cloud pressure can be as large as 200 hPa. The paper contains significant original material that can be of interest for the

developers of cloud algorithms for satellite sensors. The paper subject is appropriate to AMT. Earlier work is adequately recognized and credited. The abstract provides a sufficiently complete summary of the paper. The paper is well organized. I recommend the paper for publication after that the authors address to the following comments.

Specific comments

Line 85. Please specify the wavelength grid.

Line 90. Please explain how the synthetic radiance Raman spectrum was computed.

Lines 128-130. Provide details of retrieving cloud pressures for very small cloud fractions and for snow/ice covered areas. What values of cloud fraction are used over snow/ice? For instance, for Greenland (see Fig. 4a)?

Line 130. "In such cases, the LER method may be a good fallback". However, the authors state in Section "Scene albedo and scene pressure" that they do not recommend using scene pressures over dark areas. What could be recommended for such areas if the cloud fraction is very small?

I have also a more general question. How the LER method, i.e. scene albedo and scene pressure, can be used in trace gas algorithms?

Line 131. What ozone amounts are used in the radiative transfer simulations? Do you have nodes for ozone amounts in your lookup table? Do you include Raman scattering in your radiative transfer simulations to be consistent with the DOAS fit performed on the measured reflectances?

Lines 138-140. It would be beneficial for a reader to clarify what variables are used in the lookup table generated as a result of radiative transfer simulations. Please provide corresponding equations.

Line 145, Fig. 2. Effective cloud pressures may seem to be strange for reflectances less than 0.15 and slant column O2-O2 amounts less than 0.25*10^44. Please provide

some details of the extrapolation procedure for those values.

Line 147, Table 2. Please explain why you need so many reflectance nodes in the reflectance range of 1.0 to 2.0. Where do you get so large values of reflectance geographically?

Line 160. Can Eq. 2 be derived from the exact radiative transfer equation? If yes, please provide details in your response.

Line 163. It would be useful to provide an equation for the altitude resolved air mass factor.

Line 186, Fig.3. How large is the corresponding temperature correction factor? Please provide numbers.

Line 223, Fig. 4c. Why orbit swath footprints can be seen on the cloud fraction difference map?

Line 282, Fig. 7b. Please explain why the dependence of cloud pressure on cloud fraction is almost flat. One could expect lower cloud pressures for higher cloud fractions. Deep convective clouds in the tropics and frontal zone clouds can be examples of such cases. I think that a plot of cloud pressures vs cloud fractions in the tropics (-20<lat<20 deg) would be clarifying the issue.

Line 285. The authors state that they "have no physical explanation" for cloud pressures increasing towards lower cloud fractions. The cloud algorithm is based on an approach that treats aerosols as clouds. Low cloud fractions can be related to cloud-free scenes with tropospheric aerosols. In that case it is reasonable to anticipate higher cloud pressure retrievals. This could be a physical explanation.

Line 288. Please clarify the improvements of the interpolation scheme. It is particularly important to explain the improved performance of the scheme for low reflectance scenes.
Line 346. "... the retrieved scene pressure is significantly larger than the sea level pressure". Please characterize the difference quantitatively.

Section "Comparison with ground-based radar". It is quite desirable to compare the previous version of cloud pressure retrievals with ground-based radar data to answer a question which version of OMCLDO2 better agrees with the radar data. I think that such a question of a potential reader should be answered.

Line 392. Why not to use temperature profiles to convert cloud pressures to altitude?

Line 417. A "radar-lidar" appears in this line and elsewhere. What lidar do you mean?

Technical notes

Line 48. "the cloud pressures ... , which are sensitive near the actual cloud top". Please reword.

Line 99. Typo. Should be "all".

Fig. 2 caption. The continuum reflectance has been denoted as R in Line 85. The Greek "ro" was used for air density in Line 167.

Line 167. Greek "ro" is missing.

Line 220. Should be Table 3.

Lines 264-268. Fig. 2? Probably Fig. 3?

Line 331. Typo. Should be "pressure" instead of "fraction".

Line 175. Typo. The O2-O2 cross section appears twice in Eq. 4.
* * *

---

## Referee Comment (RC2) · Anonymous Referee #2 · 20 Jun 2016

General Comments

The paper describes several modifications to an Ozone Monitoring Instrument (OMI) cloud pressure and cloud fraction retrieval algorithm which derives information from the hyperspectral reflectance measured at the oxygen dimer absorption feature near 477 nm. The improvements to the algorithm are diverse, and two in particular, the use of climatological temperature profiles, and a new look-up table scheme, produce significant, systematic effects in the retrieved cloud pressures at small cloud fractions. Other changes have a smaller general impact but are appropriate. The authors highlight the utility of their cloud pressure data for trace gas applications, particularly where the cloud fraction is low. These new satellite cloud pressure data agree well with ground

based measurements of cloud height at mid-latitudes when cloud conditions are most favorable for comparisons. A significant amount of work has clearly gone into the updates to the algorithm which are presented here, and the focus on these low cloud fraction data is very relevant and important to the satellite trace gas community, so this work is a relevant and useful contribution to the literature.

Specific Comments

Though the authors carefully and extensively documented and investigated the changes they made in their revised algorithm, the motivation for these changes is presented less rigorously. The manuscript states that the OMCLDO2 product is "driven by what [trace gas] algorithms need for cloud information" (33), but in the introduction there is no explanation of precisely why the cloud data are needed and how they are used. The authors are asked to address this in some fashion, as it will provide context for the improvements made to the algorithm, and give greater significance to the changes and therefore this paper.

The term effective cloud fraction is used several times in the manuscript, but it is not clearly defined. A mathematical definition is easy to give in the introduction and should not be left for the reader to find in cited references. Further into the paper, the authors stress that the changes, and uncertainties, in retrieved cloud pressures are highly dependent on cloud fraction, so a definition of effective cloud fraction presented early will assist critical understanding of the reasons behind the statements. In the sentence beginning at (559), please explain what is being weighted by the effective cloud fraction.

Why are tropospheric trace gas retrievals, like those of NO2, so sensitive to pressures in particular, such that "biases will have a significant impact on trace gas retrievals, commonly limited to scenes with small cloud fractions." Is it possible to evaluate the quality of cloud pressure data retrieved at low cloud fractions in any way, given their greater associated uncertainties, and the importance the authors place on these data? The use of the scene pressure and albedo data for trace gas retrievals appears relatively limited. How are scene pressure and albedo data reported in the new product intended to be used?

The authors contrast in fig. 1 and its caption that there is a difference between the primary cloud model used to support trace gas retrievals, the Independent Pixel Approximation (IPA), and another, the Lambertian Equivalent Reflector (LER) model. With this parallel exposition it is not clear whether the authors suggest that the IPA model is, by definition, composed of Lambertian opaque reflectors. If so, this is incorrect. In fact, Zuidema and Evans (1998) which the authors cite, used a Heyney-Greenstein approximation for Mie scattering in their IPA plane parallel cloud simulations.

More likely, the authors here are aware that the IPA approach simply assumes that neighboring pixels (or satellite sub-pixels in the present case), can be described as horizontally independent with regard to radiative transfer. However the text and figure captions (parenthetical as they may be) are likely to lead some readers to think otherwise; that the IPA approach necessarily uses opaque Lambertian surfaces, and it would be unfortunate to encourage that strict association. Could the authors please clarify part of the discussion as they see fit.

The comment at (345-346), that it is common for scene pressures over the cloudless ocean to be greater than surface sea-level pressure, is puzzling. The question arises whether something, physical or numerical, similarly biases the clear sky calculations in the IPA cloud pressure estimates the authors provide?

In fig. 2 it is not clear why in the lower (cloud pressure) figure some of the circle symbols representing the table calculations near reflectance of 0.1 have a different colors than the surrounding interpolated colormap beneath. Could this be a plotting error? Also, the interpolation in the lower left of this same figure (dark red) looks somewhat strange. How far away from the table nodes is the extrapolation permitted?

Is temperature dependence of the cross-section an issue or not (179-180)?

**AMTD**

Can the statement at (235) be further explained in terms of the physics or of the model used for the cloud pressure retrievals?

Please indicate in table 3 whether the differences reported are old minus new or the opposite. It seems to be the former, consistent with maps shown in fig. 4. In table 3, the effect of new look-up tables is reported as -0.01 while at (225) this difference has a positive sign. The sign and magnitude obviously matter at low cloud fractions where a small change in cloud fraction can have a large impact on air mass factor. Please clarify.

Suggest mentioning OMI is on a satellite in the abstract near (8).

Is the amount of information in the OMI spectral range truly limited, as stated at (38)?

Replace "middle of" to "scattering within the" at (55).

Are the authors referring to differences in the algorithm, or in the results, at (72)?

Use of "cloud pressures" at (73) is unspecific. Please indicate these pressures are satellite-derived.

The correction discussed at (344) may not be needed for the OMCLDO2, but saying this about all OMI retrievals seems unwarranted.

Finally, the authors use the term "a priori" to refer to a wide range of input data used by retrieval algorithms, including theirs. For example they describe their absorption cross-section measurements and new DEM data as a priori information. From a philosophical perspective this is not entirely incorrect, but I feel the term a priori should be reserved for data that describes assumptions, often implicit, about the character of the solution to an ill-constrained inverse problem as discussed by Jackson (1979). Neither the cross-section or DEM data fall squarely within this category. The assumed shape of the vertical O2-O2 profile would however, because that information is used to constrain the inversion of O2-O2 slant column to estimate cloud pressure.

Technical Corrections - changes and additions below are in [square brackets]

(1) The readability of the title would improve if the authors changed "Improvements" to "Improvement," or replace "of" with "to."

(32) Suggest using "field-of-view" instead of "ground pixel."

(76) "DOAS (Differential [Optical] Absorption [Spectroscopy])."

(77) "fit is performed on [radiances in] the spectral region."

(83) "of the [irradiance measurement] Doppler shift."

(85) "are [matched] on the same spectral grid."

(96) "diagnostics of the fit [are] obtained."

(99) "Although [all] the information."

(99) "some [bad pixels] may remain."

(115) Remove "respectively."

(152) The phrase "even if" is ambiguous. Consider a revision.

(160) Specify n is a function of z, i.e. n(z) in eq. 2.

(175) The O2-O2 cross-section in eq. 4 is shown twice.

(349) "Comparison with ground-based [R]adar [-Lidar]."

(354) remove "As discussed below" since this is unnecessary here.

(364) Cloudnet is mentionned here for the first time here without explanation.

(437) remove the word "of."

References

Jackson, D.D., The use of a priori data to resolve non-uniqueness in linear inversion,

Geophys. J. R. Astron. Soc., 57, 137–157, 1979.

Zuidema, P. and Evans, K.F., On the validity of the independent pixel approximation for boundary layer clouds observed during ASTEX, J. Geophys. Res. Atmospheres, 103(D6), 6059–6074, 1998.

---

## Author Comment (AC1) · 23 Sep 2016

**Responses to the referee comments on the manuscript "Improvements of the OMI O2-O2 Operational Cloud Algorithm and Comparisons with Ground-Based Radar- Lidar Observations" by Veefkind et al.**

Below are the responses to the 2 anonymous referees. The comments from the referees are in **bold**, our responses in blue.

**Anonymous Referee #1**

Review of "Improvements of the OMI O2-O2 Operational Cloud Algorithm and Comparisons with Ground-Based Radar- Lidar Observations" by Veefkind et al.

General comments

This paper describes a number of improvements of the OMI operational O2-O2 cloud algorithm which provides important input data for OMI trace gas algorithms. The au- thors carry out comparisons of the new version of cloud products with the old version. They also compare the retrieved effective cloud pressures with ground-based radar data. The authors conclude that while the impact of the improvements on effective cloud fraction is small the impact on effective cloud pressure can be as large as 200 hPa. The paper contains significant original material that can be of interest for the developers of cloud algorithms for satellite sensors. The paper subject is appropriate to AMT. Earlier work is adequately recognized and credited. The abstract provides a sufficiently complete summary of the paper. The paper is well organized. I recommend the paper for publication after that the authors address to the following comments.

Specific comments

**Line 85. Please specify the wavelength grid.**

The irradiance spectrum is always brought onto the grid of the radiance spectrum. As the signal-to-noise of the solar spectrum is higher than of the radiance, interpolation in the solar spectrum is expected to yield more stable results.
In the text we have clarified that the irradiance is interpolated to the spectral grid of the radiance.

**Line 90. Please explain how the synthetic radiance Raman spectrum was computed.**

The following text was added in the manuscript

$I_R(\lambda)$ was calculated using as input the expressions given in Chance and Spurr, 1997, numbers given in Burrows et al. 1996, and the high resolution solar irradiance provided by Dobber et al. 2008. The Raman spectrum was then calculated by convolving the solar spectrum with the rotational Raman lines and the OMI slit function and divided by the convoluted solar spectrum.

Burrows, J.P., M. Vountas, H. Haug, K.V. Chance, L.C. Marquard, K. Muirhead, U. Platt, A. Richter, and V.V. Rozanov. 1996. *Study of the Ring Effect.* Techn. Report 10996/94/NL/CN, Noordwijk: ESA/ESTEC.

Chance, K., and R.J.D. Spurr,. 1997. "Ring effect studies: Rayleigh scattering, including molecular parameters for rotational Raman scattering and the Fraunhofer spectrum." *Applied Optics* 36,: 5224-5230.

**Lines 128-130. Provide details of retrieving cloud pressures for very small cloud fractions and for snow/ice covered areas. What values of cloud fraction are used over snow/ice? For instance, for Greenland (see Fig. 4a)?**

When the surface albedo that is used in the retrieval is close to 0.8, the cloud fraction becomes unstable. In the final product the cloud fraction varies between 0 and 1, however the unclipped values that are also provided in the data files show both negative cloud fractions as well as cloud fractions larger than 1. Cloud fraction larger than 1 can occur over very bright clouds, i.e. with an optical thickness larger than about 40. For these clouds the albedo is larger than 0.8. Negative cloud fractions are driven by the difference between the climatological and the actual surface reflectance.

The following was added to the text:

In case of surface albedo's close to 0.8, e.g. over snow and ice, the IPA retrieval for both cloud fraction and pressure will become unstable. An evaluation of such cases over Greenland shows that rapidly variations of the cloud fraction between 0 and 1, and variations of the cloud pressure between the surface pressure and 150 hPa.

**Line 130. "In such cases, the LER method may be a good fallback". However, the authors state in Section "Scene albedo and scene pressure" that they do not recommend using scene pressures over dark areas. What could be recommended for such areas if the cloud fraction is very small?**

Both the LER and IPA methods have problems in such cases. For LER we find scene pressure significantly larger than the surface pressure that we do not understand. This could be due to spectroscopic errors, instrumental error or model errors. Remember that because of the low scene albedo we have to make large adjustments to the scene pressure to have a significant impact on the $O_2-O_2$ absorption. Because it is hard to define a switch point between IPA and LER for the dark scenes, we would recommend to use IPA for these cases. For snow and ice covered areas we recommend using the scene albedo and pressure.

**I have also a more general question. How the LER method, i.e. scene albedo and scene pressure, can be used in trace gas algorithms?**

The use of the LER method in trace gas retrievals is relatively straightforward. Most trace gas algorithms work with look-up-tables for the airmass factors, which are tabulated as a function of amongst others the surface pressure and surface height. The LER method can be used by setting the cloud fraction to zero and using the retrieved scene albedo and scene pressure in the calculation of the airmass factor.

**Line 131. What ozone amounts are used in the radiative transfer simulations? Do you have nodes for ozone amounts in your lookup table?**

For the radiative transfer calculations we use the AFGL MLS ozone profile. As the ozone absorption optical depth is small in this spectral region and because we also fit it, the O2-O2 slant columns are not significantly impacted by ozone variations. Therefore, we do not have an entry for ozone amounts in our LUTs.

**Do you include Raman scattering in your radiative transfer simulations to be consistent with the DOAS fit performed on the measured reflectances?**

The radiative transfer calculations do not include Raman scattering.

**Lines 138-140. It would be beneficial for a reader to clarify what variables are used in the lookup table generated as a result of radiative transfer simulations. Please provide corresponding equations.**

We have added the equations to the text.

**Line 145, Fig. 2. Effective cloud pressures may seem to be strange for reflectances less than 0.15 and slant column O2-O2 amounts less than 0.25*10ˆ44. Please provide some details of the extrapolation procedure for those values.**

The text was changed as follows:

The inversion is illustrated in Fig. 2. Because the simulated spectra cover a very wide range of conditions, it is unlikely that the extrapolations in this inversion procedure have a large effect on the final result. For example, -as can be seen in lower panel of Fig. 2- the results for the effective cloud pressure around a reflectance of 0.15 and $0.4 \ 10^{-44}$ molec$^2$ cm$^{-5}$ show strange patterns due to the extrapolations. However, these combinations of continuum reflectance and $O_2\text{-}O_2$ slant columns will never occur for real atmospheres and therefore these parts of the tables are never reached.

**Line 147, Table 2. Please explain why you need so many reflectance nodes in the reflectance range of 1.0 to 2.0. Where do you get so large values of reflectance geo- graphically?**

The forward model gives very high values of the reflectance (up to 2) for extreme combinations of solar zenith angle (86°), viewing zenith angle (73°) and cloud fraction 1.2. We therefore decided to extend the dimensions to include these cases, although they will not occur often in reality. Extending the LUTs does not have a computational performance penalty.

**Line 160. Can Eq. 2 be derived from the exact radiative transfer equation? If yes, please provide details in your response.**

No, this equation is based on the assumption it that the DOAS expression, Eq. 1, is valid, which is a good approximation for weak absorption. For the wavelength range considered here absorption by O2-O2 and O3 is indeed weak. In case of stronger absorption, the slant column (i.e. the air mass factor) becomes wavelength dependent and the Ring spectrum depends on the absorption itself. We do not think that a discussion of the accuracy of the DOAS method as compared with full radiative transfer calculations is appropriate for this manuscript. Hence, we did not modify the text here.

**Line 163. It would be useful to provide an equation for the altitude resolved air mass factor.**

The following text was added in the manuscript

The altitude resolved air mass factor $m(z, \lambda)$ can be expressed as:

$$m(z, \lambda) = \frac{1}{R_0(\lambda)} \frac{\partial^2 R_0(\lambda)}{\partial k^{abs}(z,\lambda) \, \partial z} \qquad (6)$$

It represents the relative reduction in the reflectance when a unit amount of absorption is added to the atmosphere in a thin layer located between $z$ and $z+dz$. The volume absorption coefficient is given by $k^{abs}(z, \lambda) = n_{O2-O2}^2(z) \; \sigma_{O2-O2}(z, \lambda)$.

**Line 186, Fig.3. How large is the corresponding temperature correction factor? Please provide numbers.**

The concept of the correction factor is not used in the discussion of this figure. The magnitude of the correction factor can be seen in Figures 4 and 6. It ranges from 0.92 to 1.03.

**Line 223, Fig. 4c. Why orbit swath footprints can be seen on the cloud fraction difference map?**

This is an important remark and we like to thank the reviewer for pointing this out. We investigated the cross-track variation of the cloud fraction difference, which varied from approximately 0.0 at the West side of the swath to -0.01 at the East side. We found out that this was due to a shortcoming in the radiative transfer configuration that was used for generating the look-up tables. We recomputed the look-up tables with the new radiative transfer results. Fortunately, the impact on other results and conclusions is marginal. However, we did update all the figures and results in the manuscript, which obviously took some time to do. As can be seen in the updated manuscript, the East-West variation in the effective cloud fraction has disappeared. The mean difference of the cloud fraction (old minus new) is -0.01.

**Line 282, Fig. 7b. Please explain why the dependence of cloud pressure on cloud fraction is almost flat. One could expect lower cloud pressures for higher cloud fractions. Deep convective clouds in the tropics and frontal zone clouds can be examples of such cases. I think that a plot of cloud pressures vs cloud fractions in the tropics (-20<lat<20 deg) would be clarifying the issue.**

The reviewer is right that this behavior may not be flat for specific regions. To verify this, we redid Figure 7 for the latitude range -20° to 20°. As can be seen in the figure below, the highly reflective (large effective cloud fraction) clouds are at a significantly lower pressures (higher altitudes) than the clouds with low effective cloud fraction.

[Figure]

**Line 285. The authors state that they "have no physical explanation" for cloud pressures increasing towards lower cloud fractions. The cloud algorithm is based on an approach that treats aerosols as clouds. Low cloud fractions can be related to cloud- free scenes with tropospheric aerosols. In that case it is reasonable to anticipate higher cloud pressure retrievals. This could be a physical explanation.**

We believe that the effects of clouds are always dominating over the effect of aerosols. However, we checked this hypothesis by redoing the analysis of Figure 7 for the Southern ocean (lat < 40°). In this region we expect the aerosol optical depth to be very small, and cloud effects dominating of aerosols. In the difference plot (shown below) we find the same behavior as in Figure 7c. We

therefore do not attribute it to the effect of aerosols. Also the oscillation with a maximum around 0.125 Cf in the old LUTs point towards an algorithm artefact, rather than a physical effect.

[Figure]

**Line 288. Please clarify the improvements of the interpolation scheme. It is particularly important to explain the improved performance of the scheme for low reflectance scenes.**

The previous version of the LUTs was generated using a rather complicated ad-hoc method. We think that it is beyond the scope of the paper to explain the details, because we think the method we use now is significantly better and also more transparent.

We have added the following text to the section on the lok-up table inversion:

The previous version of the OMCLDO2 algorithms also made use of inverted LUTs. However, they were not calculated using radial basis functions, but computed on ad-hoc fits of the continuum reflectance and slant column $O_2$-$O_2$ versus the cloud pressure and cloud fraction. Also the number of nodes for low cloud fractions and low albedo's was significantly lower in the previous version.

**Line 346. "... the retrieved scene pressure is significantly larger than the sea level pressure". Please characterize the difference quantitatively.**

We did additional tests and changed the text to:

Over dark surface, such as oceans, the scene pressure is less well understood. For some areas over the ocean the retrieved scene pressure is significantly larger than the sea level pressure. For scene albedo's of less than 5%, about 3% of the scene pressures exceed 1050 hPa and 50% exceed 1013 hPa. We note that scene pressures larger than 1013 hPa are the results of extrapolation and therefore should be used with great caution. For dark scenes we recommend using the cloud fraction and cloud pressure, taking into account that there will be a large uncertainty in the cloud pressure in these cases (see Fig. 5).

**Section "Comparison with ground-based radar". It is quite desirable to compare the previous**

**version of cloud pressure retrievals with ground-based radar data to answer a question which version of OMCLDO2 better agrees with the radar data. I think that such a question of a potential reader should be answered.**

We expect that the correlation between the old and new OMCLDO2 data sets is much higher than between these sets and the Cloudnet product, because of the different sensitivity and the collocation errors of the satellite versus the ground based observations. Also, the largest differences between the cloud pressures of the old and new satellite data sets occur at low effective cloud fractions, which are not included in the comparison with the ground based observations, because these cases will be dominated by representation errors due to the difference in time of measurement and ground pixel size.

Below we show the original Figure 11 (left) and the same plot but for the old OMCLDO2 data set. As expected the figures look very similar, of the above mentioned reasons. For low clouds, the comparison seems to improve for the new OMCLDO2 version.

[Figure]

[Figure]

The following section was added to the text:

The comparison between the Cloudnet data was repeated for the old version of the OMCLDO2 algorithm. The results were very similar to those presented in Figure 10 and 11. This is expected because for effective cloud fractions larger than 50%, the difference between the old and the new algorithm is not very large. Moreover, the difference between the two algorithm versions is smaller than with the ground based data, because of the different sensitivity of ground based versus satellite and because of representation errors in both space and time.

**Line 392. Why not to use temperature profiles to convert cloud pressures to altitude?**

Using the scale height for the pressure to altitude conversions is just for practical reasons. We do not except that performing a more accurate conversion of pressure to temperature will have a significant impact on the conclusions. We estimate that the error that is made in the conversion is below 500 meters for almost all conditions and will be distributed quasi randomly. Such differences will not affect the conclusions that we can draw from the comparison.

**Line 417. A "radar-lidar" appears in this line and elsewhere. What lidar do you mean?**

As explained in the text, we use a Cloudnet classification product that is based on the combination of radar and lidar observations. We updated the text to consistently refer to it as radar-lidar observations.

**Technical notes**

**Line 48. "the cloud pressures ... , which are sensitive near the actual cloud top". Please reword.**

Reworded to:

This sensitivity to the middle of the clouds differs significantly from observations in the thermal infrared, which are very sensitive to the actual cloud top pressure.

**Line 99. Typo. Should be "all".**

Corrected.

**Fig. 2 caption. The continuum reflectance has been denoted as R in Line 85. The Greek "ro" was used for air density in Line 167.**

Corrected.

**Line 167. Greek "ro" is missing.**

Corrected.

**Line 220. Should be Table 3.**

Corrected.

**Lines 264-268. Fig. 2? Probably Fig. 3?**

Reviewer is right, should be Fig. 2. Corrected.

**Line 331. Typo. Should be "pressure" instead of "fraction".**

Corrected.

**Line 175. Typo. The O2-O2 cross section appears twice in Eq. 4.**

Corrected.

**Anonymous Referee #2**

General Comments

The paper describes several modifications to an Ozone Monitoring Instrument (OMI) cloud pressure and cloud fraction retrieval algorithm which derives information from the hyperspectral reflectance measured at the oxygen dimer absorption feature near 477 nm. The improvements to the algorithm are diverse, and two in particular, the use of climatological temperature profiles, and a new look-up table scheme, produce significant, systematic effects in the retrieved cloud pressures at small cloud fractions. Other changes have a smaller general impact but are appropriate. The authors highlight the utility of their cloud pressure data for trace gas applications, particularly where the cloud fraction is low. These new satellite cloud pressure data agree well with ground based measurements of cloud height at mid-latitudes when cloud conditions are most favorable for comparisons. A significant amount of work has clearly gone into the up- dates to the algorithm which are presented here, and the focus on these low cloud fraction data is very relevant and important to the satellite trace gas community, so this work is a relevant and useful contribution to the literature.

**Specific Comments**

**Though the authors carefully and extensively documented and investigated the changes they made in their revised algorithm, the motivation for these changes is presented less rigorously. The manuscript states that the OMCLDO2 product is "driven by what [trace gas] algorithms need for cloud information" (33), but in the introduction there is no explanation of precisely why the cloud data are needed and how they are used. The authors are asked to address this in some fashion, as it will provide context for the improvements made to the algorithm, and give greater significance to the changes and therefore this paper.**

The following text was added to the introduction:

The trace gas retrieval algorithms rely on information of cloud properties for each ground pixel. Clouds are very important for the radiation field in the atmosphere as they have a large impact on the paths that the photons travel from their entry at the at the top of the atmosphere, towards the detection by a satellite sensor. The photon path strongly affects the information on trace gases contained in the satellite observations. Clouds and aerosols play a double role: they shield the atmosphere below them, thus reducing the sensitivity to the trace gases in these layers, while increasing the sensitivity to layers above the clouds. In tropospheric trace gas retrievals of e.g. $NO_2$, the sensitivity of the measurement to the trace concentration as function of altitude is described by the air mass factor (e.g. Boersma et al., 2011). To compute the altitude dependent air mass factor information is needed on the cloud fraction and the cloud altitude (or pressure). A conservative estimate of the total uncertainty in the tropospheric air mass factor for $NO_2$ is estimated by (Boersma et al., 2004) as 35-60%. Uncertainty on the cloud parameters are amongst the leading errors in this estimate. Improvement on the retrieval of the cloud parameters will thus lead to a significant improvement in the tropospheric trace gas retrievals.

**The term effective cloud fraction is used several times in the manuscript, but it is not clearly defined. A mathematical definition is easy to give in the introduction and should not be left for the reader to find in cited references. Further into the paper, the authors stress that the changes, and uncertainties, in retrieved cloud pressures are highly dependent on cloud**

**fraction, so a definition of effective cloud fraction presented early will assist critical understanding of the reasons behind the statements.**

We reworded the section where the IPA and LER methods are introduced and added the definition of the effective cloud fraction as Equation 2.

**In the sentence beginning at (559), please explain what is being weighted by the effective cloud fraction.**

This was reworded to:
The top of the atmosphere reflectance is computed as the weighted average of the cloudy and clear parts, using the effective cloud fraction $c_f$ for the weighting.

**Why are tropospheric trace gas retrievals, like those of NO2, so sensitive to pressures in particular, such that "biases will have a significant impact on trace gas retrievals, commonly limited to scenes with small cloud fractions." Is it possible to evaluate the quality of cloud pressure data retrieved at low cloud fractions in any way, given their greater associated uncertainties, and the importance the authors place on these data?**

We have updated the text as follows:

As discussed in the introduction, clouds can have a shielding or an enhancing effect on sensitivity of satellite measurements of trace gases. Tropospheric trace gas retrievals are commonly limited to ground pixels with effective cloud fraction below approximately $0.2 - 0.3$, for which the cloud free reflectance dominates the scene. Figure 3 shows that for these cases the bias in the cloud pressure due to the temperature effect is very large (20-200 hPa). Such biases could change the effect of the clouds as assumed in the trace gas retrieval, from shielding to enhancing, or vice-versa, and have a significant effect on the retrieved trace gas column.

**The use of the scene pressure and albedo data for trace gas retrievals appears relatively limited. How are scene pressure and albedo data reported in the new product intended to be used?**

The scene pressure and cloud pressure retrievals are intended for scenes with snow and/or ice cover. For such scenes the cloud fraction cannot be retrieved reliably and the surface albedo estimates are often unreliable. We have implemented this retrieval for all pixels, so that users can investigate other applications as well.

**The authors contrast in fig. 1 and its caption that there is a difference between the primary cloud model used to support trace gas retrievals, the Independent Pixel Ap- proximation (IPA), and another, the Lambertian Equivalent Reflector (LER) model. With this parallel exposition it is not clear whether the authors suggest that the IPA model is, by definition, composed of Lambertian opaque reflectors. If so, this is incorrect. In fact, Zuidema and Evans (1998) which the authors cite, used a Heyney-Greenstein approximation for Mie scattering in their IPA plane parallel cloud simulations. More likely, the authors here are aware that the IPA approach simply assumes that neighboring pixels (or satellite sub-pixels in the present case), can be described as horizontally independent with regard to radiative transfer. However the text and figure captions (parenthetical as they may be) are likely to lead some readers to think otherwise; that the IPA approach necessarily uses opaque Lambertian**

**surfaces, and it would be unfortunate to encourage that strict association. Could the authors please clarify part of the discussion as they see fit.**

We have updated as follows to clarify this:

In the new version of the OMCLDO2 algorithm we use two cloud models in the radiative transfer modelling: the independent pixel approximation (IPA) (see e.g. Zuidema and Evans, 1998) and the Lambertian equivalent reflectance (LER) model. The IPA reflectance at the top of the atmosphere as the weighted average of clear and cloudy part. In our implementation of IPA, we calculate the cloudy part by treating the cloud as an opaque Lambertian reflector. For the LER method, we model the scene by assuming a Lambertian surface that covers the entire pixel. It is noted that the clouds and the ground surface in our implementation of the IPA model are also treated as opaque Lambertian reflectors. Therefore, the name LER maybe somewhat confusing, but is used for consistency with the existing literature. For each ground pixel, both the IPA and LER method is applied. The original version of the OMCLDO2 algorithm applied only the IPA method (Acarreta et al., 2004).

**The comment at (345-346), that it is common for scene pressures over the cloudless ocean to be greater than surface sea-level pressure, is puzzling. The question arises whether something, physical or numerical, similarly biases the clear sky calculations in the IPA cloud pressure estimates the authors provide?**

We agree with the reviewer that the results are puzzling. When the IPA with Lambertian clouds is applied to those scenes we will find very large error bars on the retrieved cloud pressure (see Fig 5). Moreover, for the IPA model the result will depend strongly on the error of the assumed surface albedo, that is not used for the scene pressure/albedo retrieval. In any case, the most important application of the scene pressure/albedo is for snow/ice scenes, where this problem does not occur. The reason for these overestimates have to be further analysed, but this is beyond the scope of the current paper.

**In fig. 2 it is not clear why in the lower (cloud pressure) figure some of the circle symbols representing the table calculations near reflectance of 0.1 have a different colors than the surrounding interpolated colormap beneath. Could this be a plotting error? Also, the interpolation in the lower left of this same figure (dark red) looks somewhat strange. How far away from the table nodes is the extrapolation permitted?**

We don't think that this is a plotting error. On the left side of this plot we have the points with low cloud fraction. At a cloud fraction 0 the cloud pressure can have any value but this doesn't have an effect on the slant column or the reflectance, thus all these cloud pressure values and up in the same point. The radial basis function have difficulty to cope with this point, resulting in interpolation error. However, since the uncertainty of the retrieved cloud pressure is low in this region, we consider this acceptable. In general, we do no expect large extrapolations, because we performed the radiative transfer calculations for a wide range of geophysical conditions. However, in the retrieval algorithm we cannot determine that we extrapolate, because the LUTs contain the entire field produced by the radial basis functions.

**Is temperature dependence of the cross-section an issue or not (179-180)?**

We removed the part of the sentence ".., apart from temperature dependence of the absorption cross section", as it confusing for the reader. The point that we want to make is that the effect that is described is specific for absorption by dimers such as $O_2$-$O_2$, which is independent of the temperature dependence of the cross-sections.

**Can the statement at (235) be further explained in terms of the physics or of the model used for the cloud pressure retrievals?**

This is a physical effect. The following text was added:

For small cloud fractions, the effect of the cloud on the top of atmosphere reflectance is very small, resulting in large uncertainties on the retrieved cloud pressure. In the limit of cloud-free conditions, the cloud pressure becomes undetermined. For large cloud fractions, the clouds dominate the reflectance and the cloud pressure can be determined with high precision.

**Please indicate in table 3 whether the differences reported are old minus new or the opposite. It seems to be the former, consistent with maps shown in fig. 4. In table 3, the effect of new look-up tables is reported as -0.01 while at (225) this difference has a positive sign. The sign and magnitude obviously matter at low cloud fractions where a small change in cloud fraction can have a large impact on air mass factor. Please clarify.**

Statement in line 225: For areas with low effective cloud fractions, the effective cloud fraction is approximately 0.01 higher in the version 2.

This would result in a difference of old-new of -0.01, as consistently reported in Table 3.

**Suggest mentioning OMI is on a satellite in the abstract near (8).**

Corrected.

**Is the amount of information in the OMI spectral range truly limited, as stated at (38)?**

We reworded the sentence to:
Because the amount of information on clouds in the OMI spectral range is limited, the algorithm derives an effective cloud fraction and an effective cloud pressure, instead of physical parameters.

**Replace "middle of" to "scattering within the" at (55).**

Corrected.

**Are the authors referring to differences in the algorithm, or in the results, at (72)?**

Reworded to:
In section 3 we discuss the differences in the retrieval results of the new versus the previous algorithm version.

**Use of "cloud pressures" at (73) is unspecific. Please indicate these pressures are satellite-derived.**

Reworded to:
In section 4 we present comparisons of the OMI derived cloud pressures to ground based radar-lidar observations.

**The correction discussed at (344) may not be needed for the OMCLDO2, but saying this about all OMI retrievals seems unwarranted.**

Reworded to:
A correction of the $O_2$-$O_2$ slant columns, as is sometimes used in ground based DOAS measurements (for a discussion see (Spinei et al., 2015)), is clearly not necessary for the OMCLDO2 retrievals.

**Finally, the authors use the term "a priori" to refer to a wide range of input data used by retrieval algorithms, including theirs. For example they describe their absorption cross-section measurements and new DEM data as a priori information. From a philosophical perspective this is not entirely incorrect, but I feel the term a priori should be reserved for data that describes assumptions, often implicit, about the character of the solution to an ill-constrained inverse problem as discussed by Jackson (1979). Neither the cross-section or DEM data fall squarely within this category. The assumed shape of the vertical O2-O2 profile would however, because that information is used to constrain the inversion of O2-O2 slant column to estimate cloud pressure.**

We reworded the document to avoid the term "a priori".

**Technical Corrections - changes and additions below are in [square brackets]**

**(1) The readability of the title would improve if the authors changed "Improvements" to "Improvement," or replace "of" with "to."**

We changed the title to "Improvements to .."

**(32) Suggest using "field-of-view" instead of "ground pixel."**

This was reworded to:
The $O_2$-$O_2$ cloud product (OMCLDO2) provides information on the cloud fraction and cloud pressure for each OMI observation.

*Rationale: for a push-broom imaging spectrometer like OMI "field-of-view" would refer to the entire swath width, which is 2600 km for OMI.*

**(76) "DOAS (Differential [Optical] Absorption [Spectroscopy])."**

corrected

**(77) "fit is performed on [radiances in] the spectral region."**

Reworded to:
DOAS (Differential Optical Absorption Spectroscopty) fit is performed **in** the spectral region between 460 and 490 nm …

**(83) "of the [irradiance measurement] Doppler shift."**

corrected

**(85) "are [matched] on the same spectral grid."**

Reworded to:
For each ground pixel, the irradiance ($F$) is interpolated on the spectral grid of the radiance ($I$) (see Van Geffen et al., 2015) and the reflectance is calculated as …

**(96) "diagnostics of the fit [are] obtained."**

corrected

**(99) "Although [all] the information."**

corrected

**(99) "some [bad pixels] may remain."**

corrected

**(115) Remove "respectively."**

corrected

**(152) The phrase "even if" is ambiguous. Consider a revision.**

Reworded to:
As will be described in this section, the slant column amount of $O_2$-$O_2$ depends on the temperature profile. This is not caused by a temperature dependence of the $O_2$-$O_2$ absorption cross section, but is due to the nature of the dimers, of which the absorption scales with the density, which is where the temperature comes in.

**(160) Specify n is a function of z, i.e. n(z) in eq. 2.**

Corrected.

**(175) The O2-O2 cross-section in eq. 4 is shown twice.**

corrected

**(349) "Comparison with ground-based [R]adar [-Lidar]."**

Corrected throughout the text

**(354) remove "As discussed below" since this is unnecessary here.**

Corrected.

**(364) Cloudnet is mentioned here for the first time here without explanation.**

Reworded to:
These datasets were selected because of the continuous data availability for these sites in the Cloudnet (Illingworth et al., 2007) database. Cloudnet is a network of stations for the continuous evaluation of cloud and aerosol profiles.

**(437) remove the word "of."**

Reworded to:
Updated gas absorption cross sections

**References**

Jackson, D.D., The use of a priori data to resolve non-uniqueness in linear inversion, Geophys. J. R. Astron. Soc., 57, 137–157, 1979.

Zuidema, P. and Evans, K.F., On the validity of the independent pixel approximation for boundary layer clouds observed during ASTEX, J. Geophys. Res. Atmospheres, 103(D6), 6059–6074, 1998.